# FNeVR: Neural Volume Rendering for Face Animation

**Bohan Zeng[1,†], Boyu Liu[2,†], Hong Li[1], Xuhui Liu[1,*], Jianzhuang Liu[3],**
**Dapeng Chen[4],Wei Peng[4], Baochang Zhang[1,5,*]**

[1]Institute of Artificial Intelligence, School of Automation Science and Electrical Engineering,
Beihang University, Beijing, P.R.China
[2]Sino-French Engineer School,
Beihang University, Beijing, P.R.China
[3]Huawei Noah's Ark Lab, Shenzhen, P.R.China
[4]Huawei, Shenzhen, P.R.China
[5]Zhongguancun Laboratory, Beijing, P.R.China
[*]Corresponding author, email: xhliu@buaa.edu.cn, bczhang@buaa.edu.cn
[†]Equal contributions

## Abstract

Face animation, one of the hottest topics in computer vision, has achieved a promising performance with the help of generative models. However, it remains a critical challenge to generate identity preserving and photo-realistic images due to the sophisticated motion deformation and complex facial detail modeling. To address these problems, we propose a Face Neural Volume Rendering (FNeVR) network to fully explore the potential of 2D motion warping and 3D volume rendering in a unified framework. In FNeVR, we design a 3D Face Volume Rendering (FVR) module to enhance the facial details for image rendering. Specifically, we first extract 3D information with a well-designed architecture, and then introduce an orthogonal adaptive ray-sampling module for efficient rendering. We also design a lightweight pose editor, enabling FNeVR to edit the facial pose in a simple yet effective way. Extensive experiments show that our FNeVR obtains the best overall quality and performance on widely used talking-head benchmarks. Our code is available[1].

## 1 Introduction

Aiming at animating a still head, face animation has far-reaching applications, such as photography, social media, movie production and virtual assistance, and has become a popular research topic in computer vision and computer graphics [10; 44]. It synthesizes a realistic talking-head video by combining the appearance and identity from one source face image with poses and motions extracted from a given driving video, possibly of a different person's identity from the source. With the progress of generative models, especially Generative Adversarial Networks (GANs) and Variational Auto-Encoders (VAEs), recent methods have achieved a promising performance in face animation [57; 46; 51; 12]. However, it is still a challenging task because of the difficulty of guaranteeing the motion transform validity and generating authentic images simultaneously.

Face animation methods can be mainly divided into three categories: model-free, landmark-based, and 3D structure-based. Model-free methods [55; 45; 46; 52; 7] realize compelling reenactment by modeling the relative motion field from the source to the driving faces. Notably, First Order Motion Model (FOMM) [46] significantly improves the performance of image animation by using self-supervised keypoints with 2D motion warping. Our studies show that 2D warping is effective in generating

---

[1]https://github.com/zengbohan0217/FNeVR

36th Conference on Neural Information Processing Systems (NeurIPS 2022).

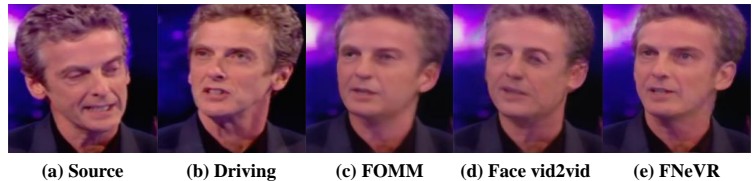

| (a) Source | (b) Driving | (c) FOMM | (d) Face vid2vid | (e) FNeVR |

Figure 1: Illustration of face synthesis results. FOMM with 2D warping produces a blurred head profile but with promising pose transformations. Face vid2vid produces a clear head profile but with poor facial details. Our FNeVR can combine their merits and generate photo-realistic images.

high-quality motion transfer. However, it has a limited ability to produce sufficiently realistic images (see Fig. 1(c)). Landmark-based methods [2; 15; 56; 49; 57] introduce 2D facial landmarks and obtain considerable progress in face animation, but they are intrinsically disadvantageous in identity preservation. Motivated by the advanced 3D Morphable Models (3DMMs) [3; 40; 16; 4; 5; 6; 31], researchers tend to integrate the 3D geometric prior to condition synthesis and achieve impressive performance to generate realistic images [30; 12]. Nonetheless, these 3D structure-based methods often focus on introducing 3D facial information in the motion model to improve the authenticity of the results, such as Face vid2vid [51], neglecting the essential advantages of 2D warping in motion transfer. As shown in Fig. 1(d), Face vid2vid produces a clear head profile but with poor facial details.

In this paper, we propose a Face Neural Volume Rendering (FNeVR) network for more realistic face animation, by taking the merits of 2D motion warping [46] on facial expression transformation and 3D volume rendering on high-quality image synthesis in a unified framework. Inspired by the remarkable properties of neural rendering [33; 22] in establishing high fidelity representations, we design a novel 3D Face Volume Rendering (FVR) module for face animation, which enables our FNeVR to learn more realistic facial details efficiently and thus achieve photo-realistic results. Specifically, the proposed FVR module first adopts two convolutional neural networks (CNNs), which we design to extract the 3D color and shape information from the 2D warped feature with the aid of the 3D reconstruction results. We only apply the 3D reconstruction module to supervise the 3D CNNs in the training process without any overhead for inference, which is different from existing 3D structure-based methods [30; 12; 51]. After that, we introduce an orthogonal adaptive ray-sampling module to efficiently calculate the sampled color field and the voxel probability for subsequent image rendering. Instead of using hierarchical sampling as in [33], our orthogonal adaptive ray-sampling only needs one MLP to process the 3D information directly. Additionally, we design a perceptual loss to guarantee the quality of rendering feature maps. All in all, our FNeVR can not only generate more realistic images than 2D-based methods, but also obtain more accurate motion transfer than 3D-based methods (see Fig. 1(e)). Moreover, we present a Lightweight Pose Editing (LPE) module, which is simple yet effective and only takes Euler angle as the input to efficiently realize facial pose editing. We further design a new loss to supervise the face editing model. The contributions of this paper include:

- A Face Neural Volume Rendering (FNeVR) network is presented to take the merits of 2D motion warping on face expression transformation and 3D volume rendering on high-quality image synthesis in a unified framework for realistic face animation.

- A Face Volume Rendering (FVR) module with orthogonal adaptive ray-sampling is proposed to capture facial details effectively and improve animation performance. Additionally, a simple yet effective Lightweight Pose Editing (LPE) module is presented only based on the Euler angle.

- Extensive experiments are conducted to compare FNeVR with state-of-the-art methods. The results show that our FNeVR obtains the best overall quality and performance on widely used talking-head benchmarks.

## 2   Related Work

**Face Animation.** Model-free methods [55; 45; 46; 52; 7; 51] complete the deformation of face images by establishing the motion field without extracting prior facial information in advance. Early models [55] can extract identity and pose information, but they may fail to capture the appropriate deformations in some problematic cases. MonkeyNet [45] handles motion transfer by acquiring

sparse keypoints. Subsequently, the First Order Motion Model (FOMM) [46] significantly improves the performance of face animation with a rigorous first-order mathematical model. While reporting promising synthesized images, most 2D model-free methods have limited ability to model out-of-plane rotations and expressional motions. Face vid2vid [51] further improves FOMM by introducing 3D representations to model a 3D motion field for face generation. Nevertheless, it has a considerable computational cost and still lacks the capability of expression transformation. Landmark-based methods [2; 15; 56; 19; 49; 57] utilize 2D facial landmarks as conditions for reenactment. However, since 2D facial landmarks cause the injection of identity-related information from the driving face to the generated one, these methods often cannot handle identity preservation well during the generation process.

In order to address the above issues, many works [28; 27; 30; 12; 41; 48] focus more on 3D structure-based methods by means of the geometric prior of 3D faces and produce impressive results on subject-agnostic face synthesis. HeadGAN [12] employs the rendered 3D mesh as input to synthesize deformed images, but it does not work well in facial expression transformation. Conditioned on the parameters of the 3D Morphable Model (3DMM) [3], StyleRig [47] and GIF [17] respectively employ pre-trained StyleGAN [25] and StyleGAN2 [26] to warp face images. PIRenderer [41] controls the face motions and predicts a flow field for deformation. Nonetheless, these methods are intrinsically limited to modeling the specific head components such as hair and teeth.

**3D Morphable Face Models.** The 3D Morphable Model (3DMM) [3] is a statistical model that provides 3D face representation for rendering face images by parameter modulation. Recent methods [40; 16; 4; 5; 6] extend 3DMM to more precise modeling of shape, expression, or pose on the basis of the original synthesis. Among these models, FLAME [31] additionally models the head pose, joint pose, and geometric expression information, offering more realistic and expressive 3D faces. Benefitting from 3DMM, the face reconstruction methods [11; 14; 1; 54] achieve significant progress. The representative method DECA [13] introduces an encoder to extract the parameters of the face for FLAME and employs another encoder to attain detailed reconstruction. In this paper, we exploit a pre-trained encoder of DECA to extract the parameters required by FLAME and then utilize FLAME to reconstruct a reliable 3D face representation from 2D images.

**3D Volume Rendering.** 3D volume rendering is used for rendering 2D projections of 3D objects and scenes. NeRF [33] is a breakthrough in view synthesis which can produce impressive rendered results based on the input positional information. Its core idea lies in optimizing a continuous volumetric scene function using a sparse set of input views. Thanks to its excellent performance, many works [43; 36; 37; 38; 8; 18; 59; 22] integrate NeRF with GANs to directly control the pose of synthesized result and generate impressive multi-view images. GRAF [43] introduces conditional GANs on the basis of NeRF essentially, which improves the quality and 3D consistency of the rendering results. GIRAFFE [38] can complete the rendering of a real-world scene and freely add multiple objects to the same scene. Differently, we introduce a lightweight and efficient adaptive volume rendering method, which only requires a single view to enhance the quality of the rendered 2D image.

## 3 Method

We formulate the face animation task as a talking-head generation with an editable pose. For a given source image $S$ and a driving video $\{D_1, D_2, \cdots, D_N\}$, where $D_i$ denotes the $i$-th frame in the video and $N$ is the total frame number, the objective of our FNeVR is to generate a photo-realistic talking-head video with the same face identity as $S$ and the motion and facial expressions derived from $\{D_1, D_2, \cdots, D_N\}$. As shown in Fig. 2, the whole framework mainly consists of four modules to achieve this goal: (1) 2D motion estimation, (2) 3D face reconstruction, (3) FVR, and (4) LPE. With $S$ and $D_i$ as input, we first utilize the 2D motion estimation module to estimate 2D facial keypoints and generate a motion field to transform the features. We also employ the 3D reconstruction module to build 3D face representations to provide the rendering network with precise and reliable 3D shape information. Then a novel FVR module is introduced to fully explore the capacity of the 2D motion module on the challenging expression transformation and generate images with high fidelity. In addition, the LPE module is added to extend our model to edit the facial pose efficiently.

### 3.1 2D Motion Estimation

Despite the trend that most existing methods [51; 21; 12] focus on studying various estimation methods of motion fields, extensive experimental results in [46; 48] show that the strategy of warping

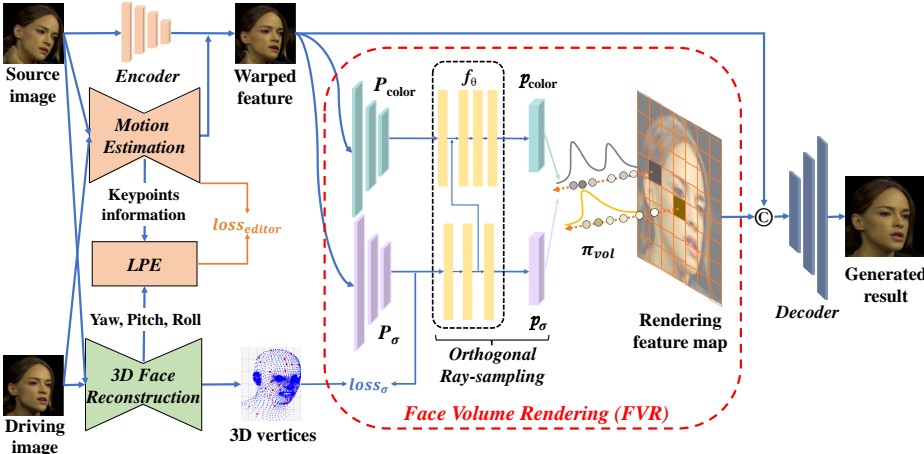

Figure 2: Overview of the proposed FNeVR, where © indicates the channel-wise concatenation. We first extract the warped feature and 3D face reconstruction representation, on which FVR is then performed to obtain the 3D shape and color information for face rendering. The 3D face reconstruction result supervises LPE and FVR by providing pose angles and 3D face vertices.

2D features achieves a consistent motion transfer of facial details. Based on this observation, we utilize the 2D motion estimation module proposed in FOMM [46] to complete the warping process. As shown in Fig. 3(a), we use an Hourglass Network [35] as the keypoint detector to extract $K = 10$ keypoints $\{p_{S,k}, p_{D,k} \in \mathbb{R}^2\}$ and their Jacobians $\{J_{S,k}, J_{D,k} \in \mathbb{R}^{2 \times 2}\}$ of both source and driving images. For each keypoint, we utilize the first-order Taylor expansion to build an affine approximation motion filed according to [46]:

$$\mathcal{T}_{S \leftarrow D,k}(z) \approx p_{S,k} + J_{S,k} J_{D,k}^{-1}(z - p_{D,k}), \tag{1}$$

where $z \in \mathbb{R}^2$ indicates a location in the image. Apart from $K$ sparse motion fields, an extra motion field is further introduced to preserve the static background information. To generate the final dense motion field, we aggregates these $K + 1$ motion fields with related weight masks $\{M_0, M_1, \cdots, M_K\}$ estimated by a U-Net used in FOMM. Furthermore, an additional occlusion map $O$ indicating regions to be inpainted is also predicted. The process of obtaining the dense motion field and the warped feature $F_w$ is:

$$\hat{\mathcal{T}}_{S \leftarrow D}(z) = M_0 z + \sum_{k=1}^{K} M_k \mathcal{T}_{S \leftarrow D,k}(z),$$
$$F_w = O \odot f_w(F_S, \hat{\mathcal{T}}_{S \leftarrow D}), \tag{2}$$

where $f_w(\cdot, \cdot)$ is the warping function, $F_S$ denotes the source feature and $\odot$ is the Hadamard product.

## 3.2  3D Face Reconstruction

As shown in Fig. 3(b), we exploit the capability of a pre-trained encoder [13] to extract the parameters required by FLAME [31] and use the prior face knowledge of FLAME to reconstruct a 3D face representation from 2D images. The reconstruction module is a typical encoder-decoder structure, where FLAME serves as the decoder, using linear blend skinning (LBS) and additional expression blend shapes to generate a detailed mesh $v \in \mathbb{R}^{3N}$ with $N = 5023$ vertices. Mathematically, given the parameters of shape $\boldsymbol{\beta} \in \mathbb{R}^{|\boldsymbol{\beta}|}$, pose $\boldsymbol{\theta} \in \mathbb{R}^{3(K+1)}$ (a global rotation vector and $K$ joints' rotation vectors), and expression $\boldsymbol{\psi} \in \mathbb{R}^{|\boldsymbol{\psi}|}$, FLAME is represented by a blend skinning function $W$ as:

$$v = W(\boldsymbol{T_p}(\boldsymbol{\beta}, \boldsymbol{\theta}, \boldsymbol{\psi}), \boldsymbol{J}(\boldsymbol{\beta}), \boldsymbol{\theta}, \boldsymbol{\mathcal{W}}), \tag{3}$$

where $\boldsymbol{T_p} \in \mathbb{R}^{3N}$ depicts a head mesh in zero pose, $\boldsymbol{J} \in \mathbb{R}^{3K}$ denotes the locations of the $K$ joints, and $\boldsymbol{\mathcal{W}} \in \mathbb{R}^{K \times N}$ acts as blend weights. The camera parameter $\boldsymbol{c}$ is further estimated to scale and translate the reconstructed vertices to the camera view. Meanwhile, the shape parameter $\boldsymbol{\beta}$ of the source image is assigned to the driving image as the reconstruction target.

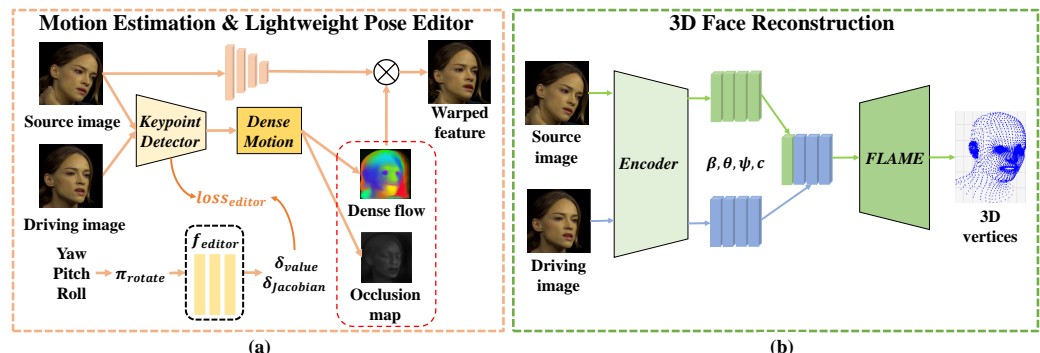

Figure 3: Illustration of Motion Estimation and LPE modules, and 3D Face Reconstruction module.

### 3.3 Face Volume Rendering (FVR)

We design a new neural rendering framework, Face Volume Rendering (FVR), for face animation as illustrated in the red box in Fig. 2. In this framework, we innovatively employ our designed CNNs to effectively extract the 3D color and shape features from the warped feature. We also introduce a new loss function to optimize the 3D shape feature with the 3D reconstruction result, which is only utilized during the training process without any overhead for inference. Moreover, we develop a new orthogonal adaptive ray-sampling module to estimate the sampled color field and voxel probability for image rendering in an efficient way. With these novel designs, our FVR module is capable of capturing more realistic facial details than 2D rendering. We also provide more details about our method in the supplementary materials.

**3D Feature Extraction.** Based on the warped feature $F_w$, we first need to extract the high dimensional 3D shape feature $F_\sigma$ and 3D color feature $F_{color}$ for subsequent face volume rendering. To this end, we respectively design two CNNs $P_\sigma$ and $P_{color}$ (see the supplementary materials), by which richer 3D shape and color features can be achieved than [33].

On the one hand, $P_\sigma$ calculates $F_\sigma = P_\sigma(F_w) \in \mathbb{R}^{H \times W \times D \times N_\sigma}$, where $D$ represents the spatial depth information and $N_\sigma$ is the number of channels representing the 3D shape information, which is set to 16 in this paper. Observing that the spatial mesh $v$ can be well reconstructed by the 3D face reconstruction module, it is thus reliable to utilize the 3D mesh feature $F_m$ derived from $v$ to supervise $P_\sigma$ to achieve a reasonable result. We maximize the inner product between vectorized $F_m$ and $F_\sigma$, which leads to a matching loss function $\mathcal{L}_\sigma$:

$$\mathcal{L}_\sigma = \exp(-\alpha_1 < F_\sigma \cdot F_m >) - \alpha_2, \tag{4}$$

where $\alpha_1$ and $\alpha_2$ are parameters to control the shape of the loss function. In practice, we empirically set $\alpha_1 = 10$ and $\alpha_2 = 0.9$. Here, we elaborate on computing $F_m$ below. Firstly, we conduct a down-sampling operation on the 3D vertices $v$ to get $N_{down}$ sampled vertices $v_{down}$, which helps to reduce the calculation cost and eliminate redundant vertex information. Additionally, we leverage the Gaussian function to process the $i$-th sampled vertex $v_{down,i}$, forming a Gaussian heatmap $F_{m,i}$ with the size of $H \times W \times D \times 1$:

$$F_{m,i}(x) = \exp(-\frac{\| x - v_{down,i} \|^2}{2\sigma}), \tag{5}$$

where $\sigma$ is the variance and we set $\sigma = 0.01$. Afterward, the 3D mesh feature of the reconstruction result $F_m$ with the size of $H \times W \times D \times 1$ is calculated by a weighted sum of all the Gaussian heatmaps $F_{m,i}$ as:

$$F_m = \sum_{i=1}^{N_{down}} w_i F_{m,i}, \tag{6}$$

where $w_i$ is the weight for $F_{m,i}$. Notably, $w_i$ is predicted adaptively by an additional CNN. It is worth noting that the 3D reconstruction vertices are not needed during inference, and we can therefore complete the image synthesis in an efficient way.

On the other hand, we need to extract the 3D color feature from the 2D warped feature $F_w$. To this end, we use $P_{color}$ to estimate the 3D color feature $F_{color}$ with the size of $H \times W \times D \times N_{color}$:

$$F_{color} = P_{color}(F_w) \in \mathbb{R}^{H \times W \times D \times N_{color}}, \tag{7}$$

where $N_{color}$ is the number of channels representing the color information.

**Orthogonal Adaptive Ray-Sampling.** With the extracted 3D features, we further calculate the sampled color field and the voxel probability, which are required for rendering. Different from conventional 3D volume rendering tasks [18; 43; 38], face animation can be considered as a single-view task, which casts rays from the camera into the scene in the direction orthogonal to the image plane. Accordingly, we design an orthogonal ray-sampling module to adaptively estimate the sampled color field $p_{color}$ and the voxel probability $p_\sigma$, which is achieved by a single MLP-based one-stage process $f_\theta$ as follows:

$$p_\sigma, p_{color} = f_\theta(F_\sigma, F_{color}) \in \mathbb{R}^{H \times W \times D \times 1} \times \mathbb{R}^{H \times W \times D \times M_{color}}, \tag{8}$$

where $M_{color}$ is the number of channels representing the color information of $p_{color}$, and the interval sampling number $D$ is the same as the previous spatial channel number $D$. Note that $p_\sigma$ is the integral of the volume density with an adaptive interval size along the orthogonal ray. In particular, the selection of the interval size is involved in estimating the voxel probability $p_\sigma$ and adaptively adjusted by the MLP.

It is worth noting that unlike the hierarchical volume sampling as in [33], which requires a two-stage network to process multiple views according to the rays of different directions, our orthogonal adaptive ray-sampling only needs one MLP to deal with the assumed ray at the angle orthogonal to the image plane. Furthermore, instead of decoupling the facial attributes as geometric-prior guided sampling methods [22], we directly process the 3D related features to predict the sampled color field and the voxel probability more efficiently. In short, by simplifying the ray-sampling with an adaptive approach, our one-stage method can significantly increase the rendering speed compared with the common two-stage process. More details about the sampling module can refer to the supplementary materials.

**Image Rendering.** Specifically, we project the sampled color field $p_{color}$ and the estimated voxel probability $p_\sigma$ onto 2D to generate the rendering feature map $F_r$. Our rendering method is a modified version of that in [32] according to orthogonal adaptive ray-sampling. Also let $i$ be a pixel point in the feature map, and $j$ be a sample interval in each pixel $i$. The rendering method calculates the final color result $F_{r,i}$ for each pixel $i$ according to the corresponding sampled color field $p_{color,i,j}$ and voxel probability $p_{\sigma,i,j}$ as:

$$F_{r,i} = \sum_{j=1}^{D} \tau_j \left(1 - \exp(-p_{\sigma,i,j})\right) p_{color,i,j}, \tag{9}$$

where $p_{\sigma,i,j}$ and $p_{color,i,j}$ represent the voxel probability and the sampled color field of the $j$-th interval of the $i$-th pixel respectively, and $\tau_j = \exp(\sum_{k=1}^{j-1} -p_{\sigma,i,k})$ is the transmittance.

Moreover, to ensure the effectiveness of the rendering feature $F_r$, we introduce a two-layer CNN $P_t$ and the VGG perceptual loss [24; 50] in the training process. Specifically, we directly input $F_r$ into the CNN to generate an intermediate image $I_m$, and then compute the perceptual loss $\mathcal{L}_R$ between $I_m$ and the driving frame $D_i$ as:

$$\mathcal{L}_R = VGG_{perceptual}(D_i, I_m). \tag{10}$$

Afterward, we concatenate the volume rendering feature map $F_r$ and the warped feature $F_w$, and feed them into our designed rendering decoder which introduces the SPADE layer [39] to obtain the final generated result. This decoder (see the supplementary materials) can enhance the rendering quality while preserving facial details of the input.

### 3.4 Lightweight Pose Editing (LPE)

Existing face animation methods, which adopt a 2D warping motion network [46; 48; 21], work well in the feature warping but fail on pose editing. As illustrated in Fig. 3(a), we introduce a lightweight network (LPE) to add a new pose editing function for our framework. Taking the rotation

Table 1: Quantitative comparison of same-identity reconstruction on VoxCeleb [34].

| Method | $\mathcal{L}_1 \downarrow$ | LPIPS $\downarrow$ | PSNR $\uparrow$ | SSIM $\uparrow$ | AKD $\downarrow$ | AED $\downarrow$ | FID $\downarrow$ |
|---|---|---|---|---|---|---|---|
| Bilayer [56] | 0.1197 | 0.4247 | 15.219 | 0.3968 | 12.60 | 0.0546 | 219.8 |
| FOMM [46] | 0.0450 | 0.1099 | 23.210 | 0.7475 | 1.383 | 0.0244 | 11.56 |
| Face vid2vid [51] | 0.0485 | 0.1051 | 22.642 | 0.7268 | 1.616 | 0.0395 | 9.142 |
| Face vid2vid-S [51] | 0.0445 | 0.0901 | 23.357 | 0.7473 | 1.421 | 0.0243 | 9.151 |
| DaGAN [21] | 0.0462 | 0.0981 | 23.263 | 0.7536 | 1.441 | 0.0247 | 9.660 |
| PIRender [41] | 0.0566 | 0.0850 | 21.040 | 0.6550 | 2.186 | 0.2245 | 11.88 |
| FNeVR (ours) | **0.0404** | **0.0804** | **24.292** | **0.7773** | **1.254** | **0.0231** | **8.443** |

angle information $\{yaw \in \mathbb{R}^1, pitch \in \mathbb{R}^1, roll \in \mathbb{R}^1\}$ provided by the 3D face reconstruction module and the keypoint information of the source image as input, LPE ($f_{editor}$) estimates the target keypoints $\delta_{value}$ and their Jacobians $\delta_{Jacobian}$. In practice, we first formulate $yaw$, $pitch$ and $roll$ as a 3D rotation matrix $r_{rotate} \in \mathbb{R}^{3 \times 3}$, and then introduce an MLP as $f_{editor}$ to compute $\delta_{value}$ and $\delta_{Jacobian}$ under a specific rotation angle $\theta_{rotate}$:

$$\delta_{value}, \delta_{Jacobian} = f_{editor}(\theta_{rotate}, p_S, J_S) \in \mathbb{R}^{K \times 2} \times \mathbb{R}^{K \times 2 \times 2}. \tag{11}$$

Moreover, we design a loss $\mathcal{L}_{editor}$ to supervise $f_{editor}$ by aligning $\delta_{value}$ and $\delta_{Jacobian}$ with the keypoints and the Jacobians of the driving frame, respectively:

$$\mathcal{L}_{editor} = \lambda_1 L_1(p_D, \delta_{value}) + \lambda_2 L_1(J_D, \delta_{Jacobian}), \tag{12}$$

where $\lambda_1$ and $\lambda_2$ are balancing parameters, and $L_1$ is the $l_1$ norm loss. Empirically we set $\lambda_1 = 1$ and $\lambda_2 = 0.5$.

### 3.5 Training

Our framework is implemented based on self-supervised learning or self-reenactment, where a pair of the source image $S$ and the driving image $D_i$ from the same speaker in a video is used as the input and $D_i$ also acts as the ground truth for supervision. We train our FNeVR by minimizing the following loss $\mathcal{L}_{total}$:

$$\mathcal{L}_{total} = \mathcal{L}_P(D_i, x_{generated}) + \mathcal{L}_G(D_i, x_{generated}) + \mathcal{L}_E(p_D, J_D) + \\ \mathcal{L}_R(D_i, I_m) + \mathcal{L}_\sigma(F_\sigma, F_m) + \mathcal{L}_{editor}(p_D, J_D, \delta_{value}, \delta_{Jacobian}), \tag{13}$$

where $x_{generated}$ is the output of the whole network FNeVR. $\mathcal{L}_{total}$ consists of four parts: (1) $\mathcal{L}_P$ and $\mathcal{L}_G$ are used to maintain the quality of the generated image; (2) $\mathcal{L}_E$ is used to adjust the 2D motion estimation module to predict reasonable motion fields; (3) $\mathcal{L}_R$ and $\mathcal{L}_\sigma$ are used to optimize the 3D volume rendering module to generate better 3D rendering features; (4) $\mathcal{L}_{editor}$ is used to optimize the face pose editor. For (1) and (2), we employ the same loss functions as in [46], including the perceptual loss $\mathcal{L}_P$, GAN loss $\mathcal{L}_G$ and equivalence constraint loss $\mathcal{L}_E$. For (3) and (4), the losses mainly provide the quality guarantee of the 3D rendering feature $F_r$.

## 4 Experiments

In addition to the extensive experiments described in this section, we also provide more results in the supplementary materials.

### 4.1 Implementation Details

**Datasets.** Our evaluation is performed on VoxCeleb [34], which contains more than 100,000 videos covering 1,251 speakers of different identities, and VoxCeleb2 [9], which contains about 1M videos of different celebrities. We adopt the same preprocessing strategy of cropping faces from the videos and resizing them to 256×256 as in [46].

**Training Details.** We train FNeVR with self reconstruction on the training set, in which a pair of source and driving images from the same speaker in a video are used as input, and the driving image also serves as ground truth for supervision. FNeVR is trained for 100 epochs, repeating the video

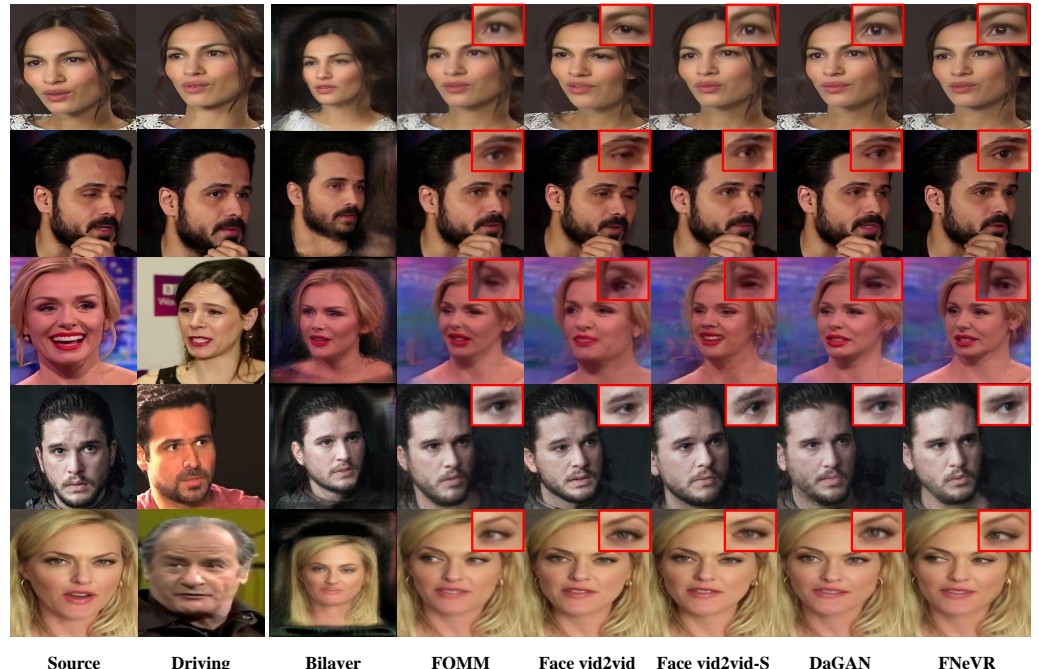

| Source | Driving | Bilayer | FOMM | Face vid2vid | Face vid2vid-S | DaGAN | FNeVR |

Figure 4: Qualitative comparison with SOTA baselines. **Top**: same-identity reconstruction; **Bottom**: cross-identity reenactment. For both tasks, our FNeVR can produce more realistic and fine-grained details than the others. For example, FOMM struggles to strike a balance between clarity and facial details, and Face vid2vid-S and DaGAN may cause distortions within some facial areas (eyes, mouth, and teeth).

Table 2: Quantitative comparision of cross-identity reenactment on VoxCeleb [34] and VoxCeleb2 [9].

| Method | VoxCeleb | | VoxCeleb2 | |
|---|---|---|---|---|
| | FID↓ | CSIM ↑ | FID↓ | CSIM ↑ |
| FOMM | 106.9 | 0.5491 | 138.1 | 0.5228 |
| Face vid2vid-S | 106.6 | **0.6447** | 148.6 | **0.6290** |
| DaGAN | 110.3 | 0.5305 | 139.6 | 0.4932 |
| FNeVR (ours) | **98.23** | 0.5505 | **133.9** | 0.5282 |

Table 3: Quantitative comparison of Flops, memory, and efficiency.

| Method | Flops(G) | Params(M) | FPS |
|---|---|---|---|
| Face vid2vid | 231.038 | 125.216 | 17.790 |
| Face vid2vid-S | 636.941 | 173.109 | 13.219 |
| DaGAN | **75.642** | 74.660 | 26.753 |
| FNeVR (ours) | 130.109 | **61.378** | **36.568** |

image set 75 times per epoch. We adopt Adam [29] optimizer with learning rate $\eta = 2 \times 10^{-4}$, $\beta_1 = 0.5$ and $\beta_2 = 0.9$ for each module. Since our model is lightweight, we only use 2 24GB NVIDIA 3090 GPUs during training.

**Evaluation Metrics.** The metrics include (1) reconstruction faithfulness using $\mathcal{L}_1$, PSNR, SSIM [53], and LPIPS [58], (2) output visual quality using FID [20], (3) Average Keypoint Distance (AKD) [46], (4) Average Euclidean Distance (AED) [46], and (5) identity preservation cosine similarity CSIM [42; 23].

## 4.2 Comparison with State-of-the-Art Methods

**Methods.** We compare our FNeVR with five state-of-the-art (SOTA) models: FOMM [46], Face vid2vid [51], Face vid2vid-S [51], Bilayer [56], DaGAN [21], and PIRenderer [41]. We use the official pre-trained models for Bilayer, FOMM, DaGAN and PIRenderer, and a widely recognized unofficial model for Face vid2vid due to the absence of the official code. Face vid2vid-S is obtained by replacing the decoder of Face vid2vid with a SPADE [39] layer-based decoder. All of these models are trained on VoxCeleb.

**Same-Identity Reconstruction.** We conduct a quantitative comparison with SOTA methods on the testing dataset of VoxCeleb when the source image and the driving image belong to the same person.

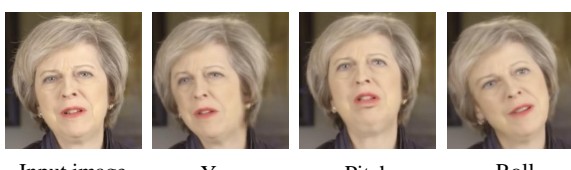

| Input image | Yaw | Pitch | Roll |

Figure 5: Visualization of the pose editing results produced by LPE.

Table 4: Ablation study for same-identity reconstruction on the VoxCeleb testing set.

| Configurations | $\mathcal{L}_1 \downarrow$ | LPIPS $\downarrow$ | PSNR $\uparrow$ | SSIM $\uparrow$ | AKD $\downarrow$ | AED $\downarrow$ | FID $\downarrow$ |
|---|---|---|---|---|---|---|---|
| FOMM baseline | 0.0450 | 0.1099 | 23.210 | 0.7475 | 1.383 | 0.0244 | 11.56 |
| FOMM + SPADE | 0.0426 | 0.0889 | 24.006 | 0.7710 | 1.316 | 0.0246 | 11.13 |
| FOMM + FVR | **0.0404** | 0.0875 | **24.374** | **0.7783** | 1.290 | 0.0233 | 10.76 |
| FNeVR w/o $F_w$ | 0.0460 | 0.0854 | 22.653 | 0.7423 | 1.367 | 0.0262 | 12.07 |
| FNeVR w/o $\mathcal{L}_\sigma$ | 0.0412 | 0.0869 | 24.163 | 0.7705 | 1.284 | 0.0239 | 9.024 |
| FNeVR (ours) | **0.0404** | **0.0804** | 24.292 | 0.7773 | **1.254** | **0.0231** | **8.443** |

As shown in Table 1, it is evident that our FNeVR outperforms other SOTA methods on all metrics. Especially, FNeVR achieves pronounced improvements on several metrics, such as image quality (PNSR), semantic consistency (AKD), and image authenticity (FID). Fig. 4(a) shows that FNeVR can generate the most realistic images that are most similar to the driving faces.

**Cross-Identity Reenactment.** We compare the performance of our FNeVR with SOTA methods on the VoxCeleb and VoxCeleb2 testing datasets when the source image and driving video come from different persons. Concretely, we randomly select 10 source images and 14 driving videos from the testing datasets. Table 2 illustrates that FNeVR produces the best overall performance. Although Face vid2vid-S has a better result in terms of CSIM, it consumes several times more computation and memory cost (FLOPs and Parameters) than FNeVR, as indicated in Table 3, demonstrating FNeVR's outstanding efficiency in compact computation. Moreover, Fig. 4(b) visualizes some synthesis results from different methods of reenactment and shows that FNeVR can generate more realistic faces whose poses and expressions are closest to the driving faces.

### 4.3 Ablation Study

We conduct comprehensive experiments on the same-identity reconstruction task to further demonstrate the effectiveness of each module in the proposed FNeVR. Specifically, we employ FOMM [46] as the baseline, and construct four comparison models: (1) replacing FOMM's decoder with our designed decoder, (2) inserting FVR into FOMM, (3) FNeVR without $F_w$ inputted into the decoder, and (4) FNeVR without $\mathcal{L}_\sigma$. The results are shown in Table 4. Both FVR and our designed decoder have significant effectiveness. In particular, FOMM with FVR inserted produces much better results than the baseline. Furthermore, FNeVR without $F_w$ performs worse than the full FNeVR, indicating that not inputting $F_w$ into the decoder will affect the model's ability to transfer the facial poses and expressions, and lead to the loss of facial details in the generated results. Furthermore, the results show that if $\mathcal{L}_\sigma$ is not introduced, all metrics are worse than the full FNeVR, indicating that it is necessary to introduce reliable 3D information. Meanwhile, even without the 3D information, FNeVR still performs well, demonstrating the great effectiveness of the proposed framework. Note that our full model produces the best overall results and surpasses other models a lot in several metrics, fully verifying the efficacy of FNeVR.

### 4.4 Lightweight Pose Editing

With the pose parameters, we conduct a qualitative experiment with pose editing visualization in Fig. 5 to further evaluate the effectiveness of the proposed LPE. Specifically, according to the Euler angle ($yaw$, $pitch$, and $roll$) provided by the 3D face reconstruction result, we separately adjust the rotation effect of the generated results. Fig. 5 shows that LPE can generate reliable face rotations according to a given Euler angle.

# 5   Conclusion

In this paper, we propose a Face Neural Volume Rendering (FNeVR) network for face animation, which unifies the 2D motion warping and 3D volume rendering in one framework. We follow the line of 2D motion warping and introduce the 3D face reconstruction module to provide reliable 3D shape information for the rendering process. We innovatively develop a Face Volume Rendering (FVR) module to enhance the facial details of the warped feature and generate high-quality faces. Moreover, we design a Lightweight Pose Editing (LPE) module, which can directly implement pose editing with rotation angles. Extensive experiments illustrate that our FNeVR achieves state-of-the-art performance.

## Acknowledgements

This work was supported by "the Fundamental Research Funds for the Central Universities", and the National Natural Science Foundation of China under Grant 62076016, Beijing Natural Science Foundation-Xiaomi Innovation Joint Fund L223024. Besides, we gratefully acknowledge the support of MindSpore[2], CANN (Compute Architecture for Neural Networks) and Ascend AI processor used for this research.

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
