# OpenReview forum: "FNeVR: Neural Volume Rendering for Face Animation"
_NeurIPS.cc/2022/Conference — NeurIPS 2022 Accept_

### Official Review · Reviewer_rGfz · 2022-07-10

**Rating:** 6
**Confidence:** 4
**Soundness:** 3 good
**Presentation:** 3 good
**Contribution:** 3 good

**Summary:**

This paper proposes a Face Neural Volume Rendering network for identity-preserving and photo-realistic face animation, which unifies the 2D motion warping and 3D volume rendering. Specifically, on top of 2d motion warping, it leverages the 3d face reconstruction for shape information and proposes a face volume rendering module to enhance facial details. It shows good quality and performance on several talking-head benchmarks.

**Questions:**

- It will be good to provide speed evaluation at inference, compared to other SOTA methods.

- Is N_\sigma equal to 1? It is a bit confusing.

- There are multiple papers that use single-pass uniform sampling when volume rendering feature maps or adaptively sample based on the geometric prior (3d face reconstruction in this case). it does not seem to be a novelty as claimed by the paper. Please clarify the contribution and do a comparison against single-stage uniform sampling or geometric-prior guided sampling.



**Ethics Review Area:**

["Inappropriate Potential Applications & Impact  (e.g., human rights concerns)"]

**Limitations:**

- As stated above, for the major contribution of the paper FVR, I do not see a strong design motivation to it given its computational cost.

- It will be good to show failure examples as well.

**Strengths And Weaknesses:**

- It well integrates 2d motion warping field, 3d face reconstruction and 3d volume rendering into a unified framework and shows the effect of each module with an ablation study. However, I think it would be good if the paper could provide more insights into the motivation to the design of each module. For example, the volume rendering is still typically expensive and why here it opts for volume rendering vs rasterization of neural texture (similar to Deferred Neural Rendering).

- Extensive experiments and evaluations. Results are quantitatively and qualitatively good. It also provides analysis why the method is inferior to SOTA on some metrics (CSIM)

- The paper is well presented.

---

> ### Author Response · Authors · 2022-08-02
> **Reply to Reviewer rGfz**
>
>
> Thank you for your positive evaluations, and we sincerely appreciate your constructive and thoughtful comments. We will address your concerns point by point below.
> ### Q1: Speed evaluation.
> + A1: Thanks for your kind suggestion. We have provided the comparison results of the inference speed in terms of FPS (Frames Per Second) in Table 3 in the revised manuscript. For the convenience of review, we also give the results here, which show that our FNeVR is faster than the state-of-the-art methods except FOMM, indicating that FNeVR not only effectively improves the generation performance, but also has an outstanding efficiency.
>   | Method         | FPS    |
>   | -------------- | ------ |
>   | FOMM           | 61.298 |
>   | Face vid2vid   | 17.790 |
>   | Face vid2vid-S | 13.219 |
>   | DaGAN          | 26.753 |
>   | FNeVR          | 36.568 |
>
> ### Q2: $N_\sigma$.
> + A2: $N_\sigma$ is equal to 16, and we have added an explanation of $N_\sigma$ to the supplementary materials. Specifically, $N_\sigma$ is the number of channels representing the 3D shape information of the 3D shape feature $F_\sigma$, where $F_\sigma$ is extracted to predict the voxel probability $p_\sigma$. As stated in Equation (8) in the paper, the fourth dimension (channel number) of $p_\sigma$ is equal to 1.
> ### Q3: Clarify the contribution and do a comparison against single-stage uniform sampling or geometric-prior guided sampling.
> + A3: Thanks for this insightful comment. We further describe this part in Section 1 and Section 3.3 of the revised paper and highlight the revised part in blue. It should be clarified that our method is quite different from the existing single-stage uniform sampling methods and geometric-prior guided sampling methods in terms of implementation.\
>   According to our knowledge, models which introduce the sampling similar to geometric-prior guided sampling include headNeRF, NeRFace, etc. The input data and model designing of these methods are completely different from our FNeVR. They sample the facial attribute information, including geometirc attributes, while our FNeVR directly samples the 3D related features. In contrast, face features are easier to deal with than face attributes, so our model uses a single MLP network to realize the target of face animation, and we also introduce the adaptive step for single-stage uniform sampling. In addition, these methods directly introduce NeRF into specific tasks, and our method uses the theory of volume rendering to design a novel decoder. It is worth mentioning that as long as enough reliable 3D information is provided to our FVR, it can also be used as a decoder for other generation tasks, such as 3D human body generation. Therefore, our method is novel and more general in comparison.
> ### Limitation1: Major contribution of the paper FVR.
> + A4: As stated in A3, we have made the motivation more explicit in Section 1 of the revised paper and highlighted the revised part in blue. The human face is a 3D object, so it is necessary to introduce 3D information in generation. Existing methods often pay little attention to the role of 3D rendering. In this paper, we design a FVR based on the volume rendering formula, which effectively improves the generation results and consumes less computational cost than Face vid2vid that uses 3D warping. It is worth noting that our FVR can be applied to more generation tasks as long as appropriate 3D shape and color information is provided. For example, if we introduce reliable 3D body information to FVR, FVR can be applied to generate photo-realistic images of the body.
>
>   The reason why we choose the volume rendering rather than rasterization of neural texture is that rasterization of neural texture still renders features at the 2D level, while our FVR transforms 2D features into 3D features, and combines the volume rendering formula to simulate the light propagation process. Besides, compared with rasterization of neural texture, FVR also considers volume density information, which can intuitively integrate spatial prior information, so as to obtain more realistic rendering results.
> ### Limitation2: Failure examples.
> + A5: As you suggested, we have provided some failure editing examples at this anonymous [**link**](https://1drv.ms/u/s!AraiW_uJqO8vhXqLwwjPf1jzC3GF?e=eQzgMw). It indicates that our LPE can generate reliable face rotations when pitch and yaw are in $[- {20^ \circ }, + {20^ \circ }]$ and roll is in $[- {25^ \circ }, + {25^ \circ }]$. However, since there are few samples of large angle changes in the training dataset, our lightweight LPE does not handle them well with a large rotation angle. This is a common challenge of many existing works such as Face vid2vid and we will try to solve this problem in the future.
>
> Thank you again for your positive comments. We hope that our rebuttal could address all your questions and concerns. Please let us know if you still have any other concerns or questions.
>
> Best regards, \
> Authors

---

### Official Review · Reviewer_n8nw · 2022-07-11

**Rating:** 5
**Confidence:** 5
**Soundness:** 3 good
**Presentation:** 2 fair
**Contribution:** 3 good

**Summary:**

This paper proposes to combine First-order motion, face vid2vid, and face volume rendering into one framework. The prior 3D knowledge from 3DMM is used to help the volume rendering procedure. The SPADE decoder from the open-sourced Face vid2vid is also used for the final prediction of the method. Experiments show that the method improves FOMM and Face vid2vid quantitatively.

**Questions:**

1. Why exactly is this paper's Orthogonal Adaptive Ray-Sampling better than the hierarchical (two-stage) volume sampling. What is the computational burden difference between the Orthogonal Adaptive Ray-Sampling and NeRF's rendering? And where is the adaptive part?

2. The authors concatenate the volume rendering feature map $F_r$ and the warped feature $F_w$ for the final prediction. What is the performance if the wrapped feature is not sent into the SPADE decoder? Is $F_w$ involved in FOMM+FVR? Is $I_m$ the results of FOMM+FVR?

3. The reviewer assumes that the Flops and memory consumption difference is led by the 3D landmark encoding procedure of Face vid2vid. Thus I am curious about the memory consumption of the models in the ablation studies (FOMM, FOMM + SPADE, FOMM+FVR). According to the paper, FNeVR would have a larger model size than these methods. Particularly, FOMM + SPADE is a combination of open-sourced code, which is the true baseline of the proposed method.

Actually, the reviewer encourages the authors to replace the SPADE decoder with a more lightweight one and adopts a variant of FOMM+FVR as the final result of this paper. As can be seen in Table 4, no obvious gain can be achieved by involving the SPADE decoder. Adding it involves more computation and makes the method less elegant.

4. The ablation study without the loss function (4) is not given. To my understanding, this is very crucial in integrating 3D information.


**Limitations:**

I have mixed feelings about this paper. Involving a simple volume rendering into FOMM like reenactment pipeline is intuitive and nice. The whole formulation, including involving 3DMM as weak prior to density estimation is not entirely novel but reasonable. Overall, I think the novelty of this paper is sufficient.

-- However, the video results of this paper are not impressive enough. Only a few videos are involved, which are not sufficient for reviewers to evaluate. No obvious improvements can be seen given the video results. Even re-training FOMM again might lead to the difference. The rotation results are also of poor quality. More importantly, no video comparisons with face vid2vid (S) are given. This is the most important weakness of this paper.

For now, I am a bit leaning towards rejection, but I will have a look at the rebuttal for the final decision. Why are the results of Face vid2vid not given? The authors should prepare a nicer video for future use of all kinds.

-- The lack of certain analysis and ablations as stated in the Questions part. Moreover, this paper should also consider comparing with other reenact works such as MARR and PIRenderer.

-- No ethical issues are discussed. As a matter of fact, FOMM itself has certain issues.

-- Certain descriptions in this paper are confusing. For example:

a) In the abstract the authors write "this is the first work to formulate the neural volume rendering of face animation with a new architecture design". To the best of the reviewer's knowledge, every work that involves neural volume rendering into face animation would somehow propose a new architecture. It is not sure what the authors are claiming.

In the introduction, the authors claim "this is the first work of neural volume rendering for face animation", which is inappropriate. NeRF has been applied to face animation in methods such as NerFACE and ADNeRF.

b) The authors use "Model-free methods" to indicate methods such as First-order Motion Model. The "Model" here might lead to misunderstandings as the authors even write "Model-free models". It is actually "human-designed intermediate structural representation" free.

-- Certain papers mentioned in the review may not be cited. Please check the references.

**Strengths And Weaknesses:**

++ The idea of involving a simple orthogonal volume rendering for face generation is actually interesting and reasonable.

++ The way of formulating the volume rendering is nice.

++ The 3D prior from 3DMM is nicely involved without interfering with the inference procedure.

++ The quantitative results outperform many previous methods.

---

> ### Author Response · Authors · 2022-08-02
> **Reply to Reviewer n8nw - Part 3**
>
> ### Limitation2: Comparing with other reenact work.
>
> **A6:**
>
> + As you suggested, we conduct more comparisons with PIRenderer and show the results in Table 1 and Table 2 of the revised paper. We use the official code and checkpoints for PIRenderer. The results are also given here:
>
>   | Method     | $\mathcal{L}_1$ $\downarrow$ | LPIPS $\downarrow$ | PSNR $\uparrow$ | SSIM $\uparrow$ | AKD $\downarrow$ | AED $\downarrow$ | FID $\downarrow$ |
>   | ---------- | ---------------------------- | ------------------ | --------------- | --------------- | ---------------- | ---------------- | ---------------- |
>   | PIRenderer | 0.0566                      | 0.1863             | 21.040          | 0.6550          | 2.186            | 0.2245           | 11.88            |
>   | FNeVR      | 0.0404                       | 0.0804             | 24.292          | 0.7773          | 1.254            | 0.0231           | 8.443            |
>
>   | Method     | FID Re1 $\downarrow$ | CSIM Re1 $\uparrow$ | FID Re2 $\downarrow$ | CSIM Re2 $\uparrow$ |
>   | ---------- | -------------------- | ------------------- | -------------------- | ------------------- |
>   | PIRenderer | 82.23               | 0.5535              | 79.3                | 0.5093              |
>   | FNeVR      | 98.23                | 0.5505              | 133.9                | 0.5282              |
>
>
>    Compared with FNeVR, the matrics of PIRenderer for reconstruction are worse. At the same time, regarding the reenactment metrics, only the image generation quality by PIRenderer has certain advantages, which is the reason for the better FID. However, according to its generated videos, the faces are blurred, and the quality of the expression transfer is poor, especially the details of the eyes and mouth. Moreover, PIRenderer suffers from a severe identity preservation problem. We also provide several video results at this [**link**](https://1drv.ms/u/s!AraiW_uJqO8vhXqLwwjPf1jzC3GF?e=eQzgMw). The main reason for the poor generation quality is that PIRenderer decouples the facial information for generation, which is ineffective compared to the methods of directly rendering warped features. The main reason for the failure of expression transfer is that PIRenderer uses pre-trained 3DMM reconstruction parameters to estimate the motion field, which loses more detailed information, making it difficult to deal with expression changes.
>
> ### Limitation3: No ethical issues are discussed. As a matter of fact, FOMM itself has certain issues.
>
> **A7:**
>
> + In fact, we have stated Social Impact in the supplementary materials, which is allowed by this conference.
>
> ### Limitation4: Certain descriptionsare confusing.
>
> + **"This is the first work of neural volume rendering for face animation."**
>
>   **A8:**
>
>   + As you suggested, we have revised the statement about "first" in the paper. However, it is worth noting that our FVR is novel and different from other methods. The only common point is that we all employ neural rendering. The basic idea of NerFACE and ADNeRF is to decouple the basic attribute information of the face, and input the basic attributes of the face into NeRF to complete the rendering. Instead of directly applying NeRF, our model deals with the single-view input to generate the driving result of a single view. We would like to emphasize that our FVR can be applied to more generation tasks as long as appropriate 3D information is provided. For example, if we introduce reliable 3D body information to FVR, FVR can be applied to generate photo-realistic images of the body.
>
> + **"Model-free methods".**
>
>   **A9:**
>
>   + Thanks for your concern. As you have understood, "Model-free methods" refer to those that do not rely on any prior knowledge of faces. We follow the existing works (such as HeadGAN and StyleHeat) to name these "Model-free methods".
>
>       In the revised paper, we have added this definition for clarity. Meanwhile, we have replaced "Model-free models" with "Model-free methods".
>
> Thank you again for your comprehensive review. We hope that this response is convincing and could address all your questions and concerns. Please let us know if you still have any other concerns or questions.
>
> Best regards, \
> Authors

---

> > ### Comment · Reviewer_n8nw · 2022-08-05
> > **Response to Rebuttal**
> >
> > Thank the authors for providing such a comprehensive reponse to all my questions. The provided video is much better than the one before. Please try involving the provided details and changes to the revision and supplementary accordingly.

---

> > > ### Author Response · Authors · 2022-08-05
> > > **Reply to Reviewer n8nw**
> > >
> > > We will definitely do it. Thank you again for your supportive comment.

---

> ### Author Response · Authors · 2022-08-02
> **Reply to Reviewer n8nw - Part 2**
>
> ### Q3: SPADE layer.
>
> **A3:**
>
> + Thank you very much for your valuable questions and suggestions. Firstly, we conduct a new ablation study (FOMM, FOMM + SPADE, FOMM+FVR) to test the computation and memory consumption, and the results are shown as follows:
>
>   |              | FLOPs(G) | Params(M) |
>   | ------------ | -------- | --------- |
>   | FOMM         | 56.24    | 59.767    |
>   | FOMM + FVR   | 72.894   | 62.413    |
>   | FOMM + SPADE | 86.799   | 56.199    |
>   | FNeVR        | 130.109  | 61.378    |
>
>   As you considered, the designed decoder which introduces the SPADE layer brings more computation consumption. Your suggestion is very reasonable, but it is worth noting that our decoder is entirely designed by ourselves and different from any SPADE decoders of other works. Although our designed decoder is not the key contribution of this paper, and the most effective improvement comes from FVR, it still helps to improve the effectiveness of the generated results, according to Table 4 of the paper. Moreover, the computation and memory consumption are far less than those of Face vid2vid, as shown in Table 3 of the paper. According to your suggestions, we will explore a lightweight decoder to achieve better results in the future.
>
> ### Q4: The ablation study without the loss function (4).
>
> **A4:**
>
> + Thanks for the insight. We have added the ablation study without the loss function $L_\sigma$ in Table 4 of the revised paper. For the convenience of review, we also give the results as follows:
>
>   | Method               | $\mathcal{L}_1$ $\downarrow$ | LPIPS $\downarrow$ | PSNR $\uparrow$ | SSIM $\uparrow$ | AKD $\downarrow$ | AED $\downarrow$ | FID $\downarrow$ |
>   | -------------------- | ---------------------------- | ------------------ | --------------- | --------------- | ---------------- | ---------------- | ---------------- |
>   | FNeVR w/o $L_\sigma$ | 0.0412                       | 0.0869             | 24.163          | 0.7705          | 1.284            | 0.0239           | 9.024            |
>   | FNeVR                | 0.0404                       | 0.0804             | 24.292          | 0.7773          | 1.254            | 0.0231           | 8.443            |
>
>   The results show that if $L_\sigma$ is not introduced, all metrics are worse than the full FNeVR, indicating that it is necessary to introduce reliable 3D information. Meanwhile, even without the 3D information, FNeVR still performs well, demonstrating the remarkable effectiveness of the proposed framework in this paper.
>
> ### Limitation1: The video results of this paper are not impressive enough.
>
> **A5:**
>
> + It is indeed our carelessness that we only provided a few comparison results of our model with FOMM and DaGAN in the supplementary materials. To be honest, we mistakenly think that providing too many videos will increase the burden of review. Hence, we now provide more video results, including the results of Face vid2vid(S) at this anonymous [**link**](https://1drv.ms/u/s!AraiW_uJqO8vhXqLwwjPf1jzC3GF?e=eQzgMw), and we emphasize the difference between our FNeVR and other methods by slowing down the video playback and zooming in some facial details to highlight it. It can be seen that the results of FOMM are relatively blurred, while the results of DaGAN have obvious defects in many frames. The results generated by our FNeVR are basically without these problems. Moreover, the results generated by Face vid2vid(S) have unnatural expressions, especially in the eyes regions, while our results are much more natural and of higher quality.
>
> + In addition, we provide more results of pose editing at this [**link**](https://1drv.ms/u/s!AraiW_uJqO8vhXqLwwjPf1jzC3GF?e=eQzgMw). According to our experiments, LPE can generate reliable face rotations when pitch and yaw are in $[- {20^ \circ }, + {20^ \circ }]$ and roll is in $[- {25^ \circ }, + {25^ \circ }]$.
>   It is worth noting that we realize pose editing in a more lightweight way, and enable 2D keypoints-based warping methods to achieve the pose editing function, such as FOMM, DaGAN, and SAFA.
>
>   All in all, the experimental results show that our model generates significantly better results with more realistic face animations. And LPE is also an innovative design. We hope the reviewer can recognize our contributions.

---

> ### Author Response · Authors · 2022-08-02
> **Reply to Reviewer n8nw - Part 1**
>
> We sincerely appreciate your comprehensive review and constructive comments. We will address your concerns point by point below, hoping that it helps you find the significance of this work.
>
> ### Q1: Orthogonal Adaptive Ray-Sampling.
>
> **A1:**
>
> + First of all, it should be noted that although both our model and NeRF use volume rendering, the purposes and implementations are different. In terms of purpose, NeRF combines neural field and volume rendering for synthesizing novel views, while our FVR leverages volume rendering to improve the quality of the warped image. Moreover, NeRF synthesizes novel images with different views according to the rays of different directions, while our FVR renders images only with the assumed ray at the angle orthogonal to the image plane, which is specially designed for face animation in this paper. In terms of implementation, our Orthogonal Adaptive Ray-Sampling does not require a two-stage network to refine the complex lights like NeRF, but only needs one MLP to process the 3D information. In short, Orthogonal Adaptive Ray-Sampling is more suitable for face animation than hierarchical (two-stage) volume sampling.
>
> + Then we explain the computational burden. The computational cost of the Orthogonal Adaptive Ray-Sampling and NeRF's rendering mainly differ in two aspects. Firstly, since NeRF and FNeVR have different generation objectives, the number of light rays inputted to NeRF is much more than the number of voxels inputted to our FNeVR, making FNeVR have less computational cost. Besides, FVR only introduces one MLP network, while NeRF needs to adopt two MLP networks as stated above, indicating that FVR is more concise.
>
> + Finally, we elaborate on the "adaptive". The adaptive part is conducted by the MLP in the FVR module. Specifically, our FVR introduces one MLP network to directly estimate the voxel probability $p_\sigma$ of each voxel which is the integral of the volume density $\sigma$ within a suitable interval size $\delta$. Therefore, for our model, the selection of the interval size $\delta$ is actually involved in the estimation of the voxel probability $p_\sigma$ and adaptively adjusted by the MLP. We have revised the analysis in Section A in the supplementary materials.
>
> ### Q2: The warped feature $F_w$ for the final prediction.
>
> **A2:**
>
> + We conduct a new ablation study when the wrapped feature $F_w$ is not sent into the designed decoder, and the results are shown as follows. We also add these results in Table 4 in the revised paper.
>
>   | Method          | $\mathcal{L}_1$ $\downarrow$ | LPIPS $\downarrow$ | PSNR $\uparrow$ | SSIM $\uparrow$ | AKD $\downarrow$ | AED $\downarrow$ | FID $\downarrow$ |
>   | --------------- | ---------------------------- | ------------------ | --------------- | --------------- | ---------------- | ---------------- | ---------------- |
>   | FNeVR w/o $F_w$ | 0.0460                       | 0.0854             | 22.653          | 0.7423          | 1.367            | 0.0262           | 12.07            |
>
>     Obviously, the overall performance by the metrics for evaluating the authenticity of the generated results is much worse, indicating that not inputting $F_w$ into the decoder will adversely affect the model's ability to transfer the facial poses and expressions, and lead to the loss of facial details in the generated results.
>
> + Yes. For FOMM+FVR, we input the concatenated results of $F_w$ and $F_r$ into the decoder just like the full FNeVR. We conduct the ablation study of FOMM+FVR to test and verify the effectiveness of our designed decoder in the paper.
>
> + As stated in Section 3.3 of the paper, $I_m$ is the output of the tiny decoder (just two layers) when its input is $F_r$. We design the perceptual loss $\mathcal{L}_R$ with $I_m$ to supervise the training process.

---

### Official Review · Reviewer_Tdy4 · 2022-07-11

**Rating:** 5
**Confidence:** 3
**Soundness:** 2 fair
**Presentation:** 2 fair
**Contribution:** 2 fair

**Summary:**

This paper proposed a unified framework FNeVR for face animation, which combines 2D motion warping and 3D volume rendering.  Specifically, a 3D Face Volume Rendering module is designed for facial details rendering. Moreover, a pose editor is also incorporated to change the facial pose. The experimental results on popular benchmarks verify the superiority of FNeVR.

**Questions:**

(1) It is better to conduct the ablation study to investigate the influence of two hyperparameters in formula (13).

(2) From the presentations (Figure 4(b)) for the cross-identity reenactment experiment, the authors only study the rendering performance when the people in source images and driving videos belong to the same gender. Do other human attributes influence the performance of FNeVR, such as gender, age, haircut, and beard?

(3) In section 4.4, the pose editor is capable of facial pose editing with a slight Euler angle. Can LPE generate reliable results when a larger Euler angle is given? Of course, the given Euler angle is reasonable for human face rotations.


**Limitations:**

In the cross-identity reenactment experiment, more experiments could be included to fully demonstrate the validity of the proposed framework, including cross-gender and cross-age reenactment experiments.

**Strengths And Weaknesses:**

The manuscript is well written and clearly clarifies the objective and main idea in the research field. The proposed framework first applies to neural volume rendering for face animation and achieves better performance than SOTA methods.

---

> ### Author Response · Authors · 2022-08-02
> **Reply to Reviewer Tdy4**
>
> We sincerely appreciate your careful and thoughtful comments. In the following we address your concerns point by point.
>
> #### Q1: It is better to conduct the ablation study to investigate the influence of two hyperparameters in formula (12).
>
> + A1：As you suggested, we utilize CSIM as the evaluation metric, which computes the cosine similarity to assess the quality of identity preservation, to provide the ablation study for the two hyperparameters of $L_{editor}$ in the following table. It shows that setting $λ_1$ to 1 and $λ_2$ to 0.5 produces the best performance. Our LPE is a separate module which does not affect the performance of the animation model. Therefore, the parameters are set empirically with the purpose of balancing the loss functions in the training process and ensuring that $L_{editor}$ does not interfere with the optimization of other loss functions.
>
>     Table 1. CSIM values with different $λ_1$ and $λ_2$ in $\mathcal{L}_{editor}$.
>
>   | Formulation | $λ_1$=0.5, $λ_2$=0.3 | $λ_1$=0.75, $λ_2$=0.4 | $λ_1$=1, $λ_2$=0.5 | $λ_1$=1.25, $λ_2$=0.6 | $λ_1$=1.5, $λ_2$=0.7 |
>   | ----------- |:--------------------:|:---------------------:|:------------------:|:---------------------:| -------------------- |
>   | CSIM        | 0.8822               | 0.8909                | **0.9051**         | 0.8973                | 0.8837               |
>
> #### Q2: From the presentations (Figure 4(b)) for the cross-identity reenactment experiment, the authors only study the rendering performance when the people in source images and driving videos belong to the same gender. Do other human attributes influence the performance of FNeVR, such as gender, age, haircut, and beard?
>
> + A2: Thanks for your kind suggestion. We have added the comparison results of cross-gender reenactment experiments in the revised paper (the last row in Figure 4(b)). More video results including cross-gender and cross-age reenactment experiments are given at this anonymous [**link**](https://1drv.ms/u/s!AraiW_uJqO8vhXqLwwjPf1jzC3GF?e=eQzgMw), and we will show more images in the supplementary materials. All of the results provided indicate that our FNeVR can achieve state-of-the-art performance in various cases. The proposed FNeVR utilizes the 2D motion estimation module to complete the warping process. It does not involve attribute decoupling, but directly performs the warping operation based on the relative motion field. Consequently, our FNeVR takes the merits of the 2D warping module and is robust to the influence of various human attributes.
>
> #### Q3: In section 4.4, the pose editor is capable of facial pose editing with a slight Euler angle. Can LPE generate reliable results when a larger Euler angle is given? Of course, the given Euler angle is reasonable for human face rotations.
>
> + A3: Please note that we already provided a demo video in the supplementary materials to show the editing results under different Euler angles, and we also put more edited videos at this [**link**](https://1drv.ms/u/s!AraiW_uJqO8vhXqLwwjPf1jzC3GF?e=eQzgMw). According to our experiments, LPE can generate reliable face rotations when pitch and yaw are in $[- {20^ \circ }, + {20^ \circ }]$ and roll is in $[- {25^ \circ }, + {25^ \circ }]$. However, as you concerned, since there are few samples of large angle changes in the training dataset, our lightweight LPE does not handle them well with a large rotation angle, resulting in a certain degree of distortion. This is a common challenge of many existing works such as Face vid2vid and we will try to solve this problem in the future. As stated in the paper, our LPE is based on a 2D keypoint warping framework which does not require any 3D face geometric prior during the inference process, while existing works usually achieve pose editing with the help of a 3D face prior, which increases the computational cost of the network. Our LPE can be conveniently inserted into existing 2D keypoint warping frameworks. To the best of our knowledge, no face animation works based on the 2D keypoint warping framework have the pose editing function (such as FOMM, SAFA and DaGAN), and our LPE is the first one. Moreover, compared to 3D-based works, our advantage lies in using a lighter model to generate sufficiently realistic results.
>
>   Finally, we would like to emphasize that the main contributions of this work are the novelty and effectiveness of the FVR module, with LPE as a complement. As stated in A2 above, our method can produce excellent performance in animating still face images in various cases. We have revised our paper to emphasize the contributions.
>
> Thank you again for your review. We hope that our response helps to address all your concerns. Please let us know if you still have any other concerns or questions.
>
> Best regards, \
> Authors

---

### Author Response · Authors · 2022-08-02
**The Response Letter for  Paper3218**

We sincerely thank the editor and all reviewers for the valuable suggestions and inspiring criticisms. We believe that all comments have been carefully accommodated to the best of our knowledge. Newly added or modified texts are highlighted in blue in the revised manuscript. According to the reviewers, we have emphatically revised the motivation of the proposed FNeVR and the details of orthogonal adaptive ray-sampling, and added more ablation studies about the effectiveness of the wrapped feature $F_w$ and $L_\sigma$. Moreover, in order to present the impressive performance of our FNeVR, we have provided more video results at this anonymous [**link**](https://1drv.ms/u/s!AraiW_uJqO8vhXqLwwjPf1jzC3GF?e=eQzgMw).

Furthermore, we would like to emphasize the main contributions of this work:
+ We present a Face Neural Volume Rendering (FNeVR) network which takes the merits of 2D motion warping on face expression transformation and 3D volume rendering on high-quality image synthesis in a unified framework. FNeVR can not only generate more realistic images than 2D-based methods, but also obtain more accurate motion transfer than 3D-based methods.
+ We propose a Face Volume Rendering (FVR) module with orthogonal adaptive ray-sampling to capture facial details effectively and improve the animation performance. Unlike NeRF and its related works, our FVR directly processes the 3D features and efficiently generates the driving result of a single view.

We greatly appreciate the editor and all reviewers for your time in reviewing our revised manuscript and response.

Best regards, \
Authors

---

### Meta-Review · Area_Chair_xE8a · 2022-08-30

**Recommendation:** Accept
**Confidence:** Certain

**Metareview:**

After rebuttal all reviewers recommend acceptance. The authors are encouraged to follow the reviewer suggestions on improving the final paper.

**Award:**

No

---

### Decision · Program_Chairs · 2022-09-14

Accept